# A Coherent on Receive X-Band Marine Radar for Ocean Observations

**DOI:** 10.3390/s21237828

**Published:** 2021-11-25

**Authors:** Jochen Horstmann, Jan Bödewadt, Ruben Carrasco, Marius Cysewski, Jörg Seemann, Michael Streβer

**Affiliations:** Helmholtz-Zentrum Hereon, Max Planck Str.1, 21502 Geesthacht, Germany; Jan.Boedewadt@hereon.de (J.B.); Ruben.Carrasco@hereon.de (R.C.); Marius.Cysewski@hereon.de (M.C.); Joerg.Seemann@hereon.de (J.S.); michael.Stresser@hereon.de (M.S.)

**Keywords:** marine radar, coherent radar, surface waves, grazing incidence, Doppler radar, coherent on receive radar

## Abstract

Marine radars are increasingly popular for monitoring meteorological and oceanographic parameters such as ocean surface wind, waves and currents as well as bathymetry and shorelines. Within this paper a coherent on receive marine radar is introduced, which is based on an incoherent off the shelf pulsed X-band radar. The main concept of the coherentization is based on the coherent on receive principle, where the coherence is achieved by measuring the phase of the transmitted pulse from a leak in the radar circulator, which then serves as a reference phase for the transmitted pulse. The Doppler shift frequency can be computed from two consecutive pulse-pairs in the time domain or from the first moment of the Doppler spectrum inferred by means of a short time Fast Fourier Transform. From the Doppler shift frequencies, radial speed maps of the backscatter of the ocean surface are retrieved. The resulting backscatter intensity and Doppler speed maps are presented for horizontal as well as vertical polarization, and discussed with respect to meteorological and oceanographic applications.

## 1. Introduction

The marine radar (MR) was developed for navigational purposes in the shipping industry to improve safety during nights or bad weather conditions. However, MRs have also been shown to be extremely useful for remote sensing of various hydrographic parameters such as wind, waves, currents as well as changes in bathymetry and shoreline [1,2].

The classical MR operates in X-band (9.4 GHz), at grazing incidence, as an incoherent pulsed radar system transmitting a short (typically 50 to 100 ns) electromagnetic pulse and measuring the returned backscatter intensity in space (in polar coordinates) and time, which is related to the radar cross section (RCS). A MR is a typical imaging radar that extracts a RCS image of its surroundings (approximately every 2 s) by scanning the area with its rotating antenna. With respect to hydrographic applications, the RCS of the ocean surface primarily results from the small-scale surface roughness (~3 cm). At grazing incidence (>85°), the RCS is proportional to the spectral density of the surface roughness on scales comparable to half the radar wavelength (Bragg scattering), which in the case of X-band is ~1.5 cm. In addition, at grazing incidence, and especially at horizontal polarization, the radar backscatter results from other scattering mechanisms, e.g., small-scale wave breaking and wedge scattering (for details refer to the special issue by Brown [3]). The MRs for hydrographic application’s record this information in time and space digitally, enabling the retrieval of various parameters such as wind [4,5], waves [6,7] and currents [8,9].

In contrast to incoherent MRs, coherent systems transmit the electromagnetic signal with a known phase. In addition to the intensity, they are able to measure the phase of the incoming signal and thus to retrieve the Doppler speed. For this reason, they are often referred to as Doppler radars. Coherent MRs operating in X-band at grazing incidence are expensive and have not been widely used in meteorological and oceanographic (METOC) applications so far. However, there is a lot of potential for coherent MRs in particular with respect to current [10,11,12] and wave [7,13,14,15] retrieval. Note that non imaging X-band Doppler radars are being used operationally for detection of rain and there are several other METOC applications with regards to storms, gusts and other important weather phenomena (refer to Keeler and Serafin [16]).

Within this paper, a coherent on receive MR is introduced, which is based on an incoherent, off the shelf X-band radar system. In contrast to fully coherent systems, incoherent magnetron based radars transmit pulses of random unknown phase. To coherentizise the system, the main concept is based on the coherent on receive principle [17]. Therefore, the phase of the transmitted pulse is measured from a leak in the radar circulator, which serves as a reference phase for the transmitted pulse. The Doppler shift frequency can be computed from two consecutive pulse-pairs in the time domain [18,19] or from the first moment of the Doppler spectrum inferred by a Fast Fourier Transform (FFT) of a short time series of pulses [20,21]. In 2000, Hatten et al. [10] introduced one of the first developments of these systems for surface current retrievals. Later this system was described in detail showing preliminary results of Doppler dependencies on wind and currents [22]. In 2010, Trizna [13] utilized a similar system to investigate surface currents and wave spectra. A detailed description of their system is given in [23]. In contrast to the above-mentioned systems the coherent on receive MR of the Helmholtz-Zentrum Geesthacht (HZG) has the potential to operate with a rotating or static antenna as well as to operate at horizontal or vertical polarization. This system has already been utilized for observation of surface currents [11,12] and surface waves [7,24,25,26]. However, a detailed description of HZG’s MR system with the involved tuning and processing steps as well as with intensity and Doppler measurement examples for different METOC applications, has not been published so far.

Within Section 2, the radar hardware and its modification for the needs of coherentization and its METOC applications are introduced. In Section 3, the different tuning and processing steps are described to acquire high quality intensity and Doppler speed data from HZG’s MR. Typical intensity and Doppler speed measurements, using different polarizations, are presented for deep-water and coastal stations including a further example in the marginal ice zone (Section 4). Finally, in Section 5, conclusions and outlook are given.

## 2. Description of Radar Hardware

HZG’s coherent on receive MR system is based on an off the shelf marine radar scanner and an antenna (Figure 1). The radar scanner was modified by adding the following components: linear amplifier, analogue to digital converter (ADC), control board and a step motor with an angular position transmitter. The radar is controlled by a computer (Figure 2).

Since 2011, the radar system is based on the radar scanner SU70-14E from GEM electronica, which operates in X-band (9.4 GHz) with a peak power of 12 kW. For METOC applications the radar is operated with a short pulse (50 ns) representing a range resolution of 7.5 m and a pulse repetition frequency of up to 2 kHz. The radar system is operated with a dedicated software with a graphical user interface running on a standard Windows-PC that is also used to store and process the acquired data. The internal control of the scanner unit is handled by the original communication board as well as a custom made control board designed in close collaboration with St. Petersburg Electrotechnical University, Russia. The communication board gets its information from the PC as well as the main control board. It controls the warm up cycle, switching between transmission and standby as well as the tuning of the modulator. Furthermore, it provides the PC and control board with information on the power to the magnetron and the coarse tuning quality. The control board triggers the modulator and ADC. It also converts the incoming stream of every pulse from the ADC into a complex numbered string, which is sent via LAN to the PC for storage of the data. In addition, the control board controls the motor responsible for the azimuthal pointing of the antenna. Two different antenna operations have been implemented, the static and rotation modes. In the static mode the antenna is oriented in a pre-selected fixed direction. In the rotation mode the antenna performs 360° azimuthal scans of the ocean surface at a rotation speed of 30 rpm. In Table 1 the most important parameters of the unit with respect to METOC applications are listed.

The scanner operates on the transmitter side with a fully solid state modulator and a magnetron that generates the pulse sending it through the circulator to the antenna for transmission. A small portion of the pulse leaks through the circulator and runs through the radar front end and is digitized by the ADC representing the reference phase of the transmitted pulse.

The radar can be operated with either horizontal (HH) or vertical (VV) polarization antennas. The antennas are slotted waveguides with a frequency band of 9410 MHz ± 50 MHz and are available in several lengths (4′, 7.5′ and 9′). In Table 2 the main characteristics of the different antennas are summarized.

The backscattered signal is received by the antenna and runs through the circulator to the radar front end where the signal passes through the frequency mixer. The output of the frequency mixer has an intermediate frequency of 60 MHz, which is typical for this type of radar. From there, the analog signal takes two paths. The first path goes directly to the first channel of the ADC where the signal is sampled at 80 MHz at 14 bit in in-phase (I) and quadrature (Q) channels, which are 90° out of phase. The second path goes to a custom designed super low noise linear amplifier. The amplifier consists of a three step transistor amplifier, where each step has an amplification factor of approximately 10, resulting in a total amplification of approximately 40 dB. From the amplifier the signal runs into the second channel of the ADC and is sampled in the same manner as the first. The receiver provides a sensitivity of $P_{min}$ = −120 dBW [22], where due to the amplified and non amplified path, the linear bandwidth covers more than 100 dB. Within the control board both streams are downsampled (by averaging) to 20 MHz and collected in a series of 435 bins which are sent via a 100 Mbit LAN to the PC for final storage on hard disks. This process is performed for every individual pulse with a pulse repetition frequency of up to 2 kHz. In this configuration the system generates about 211 Megabytes of raw radar data (binary) per minute.

## 3. Signal Tuning and Processing

There are several steps required prior to acquiring radar data for METOC applications. These include the adjustment of the radar pulse length, the estimation of the MR internal time delay between transmission of a pulse and start of the ADC and the fine-tuning of the intermediate frequency. After these procedures have been performed, the data can be processed to backscattered intensity and Doppler speed, where the latter can be achieved by the pulse pair or spectral method.

### 3.1. Adjustment of Pulse Length

The adjustment of the pulse length is performed in the laboratory. Using an oscilloscope, the pulse length is manually adjusted to approximately 50 to 70 ns under consideration of an acceptable minimum power and a reasonable pulse shape. The typical pulse of an adjusted radar is depicted in Figure 3. The yellow line depicts the pulse, while the pink line is the intermediate frequency of the signal at the front end. The pulse shape differs significantly from the theoretical shape (rectangular pulse). This can lead to significant misinterpretation of the measured signal and should be taken into account when analyzing the data [27].

### 3.2. Estimation of Internal Time Delay

Due to the construction of the radar, there is a time delay between the transmission of the pulse and the start of acquisition of the ADC. To estimate this delay, the first digitized bins (where the transmitted signal is stored) of a series of individual pulses (≥256) are investigated with respect to finding the bin with the mean maximum intensity. This bin contains most of the energy of the transmitted pulse and therefore the best phase information. This procedure is performed in post processing as the bin number can differ between radar systems. In Figure 4 an example of the digitized bins 13 to 20 are plotted for 29 pulses, which result from the transmitted pulse leaked through the circulator. In this case the bin with the maximum mean amplitude is bin 16 (left hand side), so that the reference phase of the transmitted pulse is at bin 16 (right hand side).

### 3.3. Fine Tuning of the Intermediate Frequency

To get the cleanest possible signal, the gunn diode within the radar front end has to be fine tuned with regards to the intermediate frequency (IF). The IF must be tuned to the radar internal resonance frequency. To perform this tuning the I and Q-channel of the digitized transmitted pulse is investigated with respect to the I Q balance [28]. The ideal tuning is reached when the I and Q channels are located on a perfect circle on the I Q scatterplot (Figure 5c), which means that the amplitudes in the I and Q channels are identical, and both channels are shifted by 90°. Tuning the IF off the radar internal resonance frequency, generates an extra phase shift, producing an elliptical shape on the I Q scatterplot (Figure 5a,b).

Unfortunately, the IF can change slightly over time, which is most likely caused by the temperature variation within the radar scanner. Therefore, adjustments may have to be performed in case of large temperature changes. In the future, this IF tuning will be implemented as a fully automated process within the radar.

### 3.4. Backscatter Intensity and Doppler Speed

The radar backscatter intensity results from the I and Q channels and is given by
(1)$$A^{2}=I^{2}+Q^{2}$$

The Doppler speed can be determined by two different methods: the so-called pulse pair method [18,19] and the spectral method [20,21].

With the pulse-pair method, the Doppler shift frequency $f_{D}$ is computed using the time derivative of the instantaneous phase $\varphi_{el}$:(2)$$f_{D}=\frac{1}{2}\pi \left(\frac{d\varphi_{el}}{dt}\right)=\frac{1}{2}\pi \;d\varphi_{el}\;PRF$$
where, *t* is time, and *PRF* is the pulse repetition frequency of the radar. The corresponding Doppler speeds are obtained by
(3)$$U_{D}=\frac{\lambda_{el}}{2}f_{D}$$
where $\lambda_{el}$ is the electromagnetic wavelength of the radar. Note that the Doppler speed represents the line-of-sight component of the scatterer motion. Thus, this projection effect should be considered for most METOC applications. To reduce the noise of the calculated Doppler speeds, the frequency shifts of all pulses that fall within a predefined grid are averaged.

For the spectral method a number (ensemble) of consecutive radar pulses has to be analyzed. A good choice of an ensemble size strongly depends on the application however, ensemble sizes of 1024 samples (corresponding to 0.512 s at a *PRF* of 2 kHz) have shown excellent results [26,27]. To obtain the Doppler spectrum $S_{D}\left(f\right)$, each ensemble is transformed to the spectral domain using a Fast Fourier Transform (FFT). The first moment of the Doppler spectrum defines the Doppler shift frequency
(4)$$f_{D}=\frac{\int^{\text{​}}fS_{D}\left(f\right)}{\int^{\text{​}}S_{D}\left(f\right)}$$
which can be directly transformed to the Doppler shift velocity using Equation (3).

Both methods have their pros and cons, which are discussed in Hwang et al. [14].

To get an estimate of the accuracy of the Doppler speeds retrieved from the MR, several measurements of a corner reflector mounted on the beach in a distance of ~1200 m were acquired. The mean Doppler speed of 50 pulses resulted in a root mean square error of approximately 0.02 m/s.

To get a measure of the ensemble variability of the Doppler speeds retrieved from the MR, a confidence parameter is defined [7], which is given by the alignment of the vectors pointing to the phase difference in the complex plane. The confidence is
(5)$$conf=\frac{\left|\sum_{j=1}^{511}C_{i}\right|}{\sum_{j=1}^{511}\left|C_{i}\right|}\;\mathrm{with}\;C_{i}=\frac{A_{j=1}e_{j+1}^{i\phi_{j+1}}}{e_{j}^{i\phi_{j}}}\;i,j\;\mathrm{element}\;\mathrm{of}\;\left[1,\;n-1\right]\;$$
where *A* is the amplitude, ϕ the phase of the vector and *n* the number of pulses within the ensemble. If the confidence parameter is high, this means there is a low variability of the Doppler speeds within the ensemble. The confidence parameter has a different physical meaning under various situations, as will be discussed below.

Figure 6 shows an example Doppler spectrum and the corresponding polar representation of the complex pulse pairs ($C_{i}$ in Equation (5)) computed for an ensemble of 1024 radar pulses within one range cell at r = 262.5 m. The Doppler shift frequency obtained with the spectral method (solid black), and the pulse-pair method (dashed magenta) are very similar for this particular ensemble. Also shown is the standard deviation (i.e., the second moment) of the shown Doppler spectrum, which is also a measure for the velocity variability within the radar range cell during the integration time of the ensemble.

## 4. Measurement Results

In the following, measurements from two permanent radar stations in the German Bight of the North Sea are shown. The first radar station is on the offshore research platform FINO-3, which is located 80 km west of the island Sylt in the German Bight of the North Sea. FINO-3 is standing at a water depth of 22 m, which increases over a distance of 3 km towards the northwest to 27 m. The radar was installed on the northwest side of the mast at a height of 43 m over mean sea level. FINO-3 is located in open waters with prevailing wind and waves from the west and circulating tidal currents of up to 0.5 m/s.

The first example of MR data from FINO-3 was acquired at HH-polarization in the rotational mode. The significant wave height was 5.5 m with a peak wave direction of 312° and a peak wave period of 13.3 s, as measured by a wave rider buoy located within 300 m of the platform. The mean wind speed was 18.6 m/s coming from 338° and was measured by an anemometer at 30 m height. Figure 7 shows the intensity and Doppler speed images from a single radar antenna rotation from 29. October 2017 at 11:54 UTC. Due to the very low number of pulses available for each resolution cell the Doppler speeds were retrieved via the pulse pair method, with 32 pulses per ensemble. In the southeast of the images the radar signal is strongly disturbed by the metal structure of the mast of the FINO-3. In the northeast of the intensity image several bright targets are visible (refer to white circles in Figure 7 left hand side), which result from the individual wind turbines from the offshore wind park Dantysk. In both the intensity and Doppler images a wave pattern can be seen, which is propagating through the imaged area from the northwest. In the intensity image there is a clear decrease of the mean radar backscatter intensity over the range, which is due to the range spreading loss of the radar signal and the effect of tilt modulation [29]. Furthermore, an azimuthal dependency of the radar backscatter is observed, which is due to the radar look direction with respect to the wind direction, often called wind direction dependency of the radar backscatter [5,30,31]. A further artifact in the intensity data, is the effect of wave shadowing. With increasing range (increasing grazing angle) waves shadow parts of the following waves, which lead to a significant reduction of backscattering [32]. In case of the radar utilized here the returned signal in these shadowed regions is below the noise floor of the radar and therefore results in no detection of backscatter signal in these areas [7].

In the Doppler speed image (Figure 7, right) the mean Doppler speed is strongly changing with azimuth angle, where the maximum Doppler speeds are observed in up wind direction (320°) and the lowest in down wind directions [11,22,31]. In addition, there is a slight increase of the mean Doppler speed with range, which is most likely due to shadowing of sub resolution surface waves [7,25]. Furthermore, in the shadowed areas of the waves the phase is very noisy; this is due to the very low backscatter, which is below the noise floor of the system and therefore results in a random phase and a low confidence level. Also the wave structures are clearly visible in the Doppler data, which are caused by the wave induced orbital motion that in turn leads to the movement of the surface scatterers at different speeds with respect to the location on the waves [7,14].

Figure 8 shows radar data, which were acquired with the same radar 19 min after the data shown in Figure 7, however, in the static antenna mode. In this case the radar antenna was oriented with a 315° angle looking into the peak wave direction. Here, the intensity and Doppler speed were computed using the pulse-pair method with 1000 individual pulses per ensemble, representing the mean intensity and Doppler speed over 0.5 s. Due to the large ensembles this processing results in a significantly noise reduced measurement of the intensity and Doppler speed. Again, the decrease of radar backscatter intensity over range can be observed as well as the strong modulation of the intensity due to the waves. In addition, the increase of shadowed areas with range distance can be observed. Furthermore, the resulting Doppler speeds (Figure 8 center panel) are not as noisy as in the rotational data, which is due to the significantly larger ensemble size. To remove the random phases resulting from the very low backscatter in the shadowed areas, a filter was applied, that masks all areas with a confidence level (Equation (5)) below 0.5 (white masked area in Figure 8 right hand side). The slope of the wave pattern gives the propagation speed of the dominant waves along the look direction of the radar. The higher Doppler speeds (>7.0 m/s) of some of the waves are due to wave breaking, where Doppler speeds peak up to the phase velocity of the waves [33,34,35]. Due to the very strong wind blowing towards the radar, the mean wind induced Doppler speed is higher than the negative contribution to the Doppler speeds induced by the wave orbital motion. For this reason, there are no negative Doppler speeds in Figure 8.

These data have been used to estimate the significant wave height from the Doppler speeds by relating them to the orbital velocity of surface waves and retrieving the significant wave height via linear wave theory with an accuracy of 0.21 m without any calibration parameters [7].

Another radar station is located on the west coast of the island Sylt on a cliff in the vicinity of the beach with a distance of ~50 m to the mean water line, and water depths increasing up to 10 m towards the west in a distance of 3 km from radar. The radar is mounted on a mast at a height of 28 m over the mean water level. The west coast of Sylt has an intermediate tidal sandy beach where predominant wind and waves are from the west and the tidal currents are along shore. On this part of the island bathymetry as well as the beach undergo strong changes during storms and are reconstructed frequently by beach nourishments.

At the Sylt station, the radar was operated in the static antenna mode with a HH-polarized antenna, which was changed after one hour to a VV-polarized antenna. During this time period the weather situation was fairly constant with wind speeds of approximately 16 m/s from the west (measured at the station) and waves coming from the west with a peak period of 8 s and a significant wave height of approximately 2.7 m (measured by a directional waverider 1.3 km west of the station). Figure 9 show the Intensity for HH- and VV-polarization respectively. It can be seen that the radar backscatter intensity of the VV-polarization is slightly higher than HH-polarization, which is well known for moderate incidence angles [36]. Furthermore, the waves are imaged with much more clarity in the VV-polarization, which is due to the scattering mechanisms, as HH-polarization is more strongly influenced by wave breaking. The differences between HH- and VV-polarization are significant in the case of the Doppler speeds. The mean Doppler speed is much larger in the HH-polarized data, which is due to the higher sensitivity of HH-polarization to wave breaking [37,38]. Therefore, the Doppler speeds in VV-polarization are much better suited for retrieving the orbital motions of waves or mean surface currents.

Breaking waves have clear signatures for both, HH and VV. They show up as regions with significantly increased intensity and Doppler velocity. However, the Doppler signal must be carefully interpreted, since the presence of steep and breaking waves can cause artifacts at grazing incidence due to radar pulse smearing [27]. Nonetheless, if this is considered, the spatial increase of the Doppler velocity can be related to roller energy and wave dissipation [26,27].

A further example of HZG’s MR data was collected by the RV Lance along a field of sea ice in the Fram Strait on the 13. September 2015 at 14:00 UTC. The mean amplitude of 50 rotations of MR data are depicted in Figure 10a and the confidence calculated from Equation (5) in Figure 10b. At the acquisition time the ocean surface was very calm and no waves were imaged by the radar, which makes the distinction between sea ice and open water from the radar backscatter fairly easy [39]. This can also be achieved by looking into the confidence, which is a measure of the variation of the Doppler speed and therefore can be utilized to detect the speed fluctuations over open water on short time scales (rotation to rotation) in comparison to those over the sea ice.

## 5. Conclusions and Outlook

A coherent on receive MR system has been developed for the measurement of intensity and speed of the ocean surface backscatter in space and time. The system is based on an off the shelf pulsed MR, which was modified to measure the outgoing and incoming phase to retrieve the Doppler speed. The radar can be operated with a rotating antenna, as well as with a fixed orientation of the antenna. The latter enables the acquisition of significantly noise-reduced Doppler speed data. Furthermore, the radar can be operated with either HH- or VV polarized antennas. In particular for the Doppler based wave and current applications VV-polarization is the preferred polarization. The radar can be operated from offshore platforms, onshore stations as well as marine vessels. A number of applications have been developed over the last decades to retrieve bathymetry, surface currents, wind and waves from the intensity data. In the case of the radar retrieved Doppler speeds, the main applications are focused on surface currents as well as wave properties. However, there is also some potential with respect to sea ice, like distinguishing between open water and sea ice.

In the future, the radar will be expanded to sample up to 2048 bins, which will significantly extend the range to over 14 km, which is of particular interest for coastal stations. In addition, an automated fine-tuning to the intermediate frequency will be implemented to accommodate for drifts during long-term operation. Furthermore, the I and Q channels will be oversampled with 80 MHz to investigate possible improvements of the Doppler speed measurements, which would be particularly important for the rotating mode of the antenna.

## Figures and Tables

**Figure 1 sensors-21-07828-f001:**
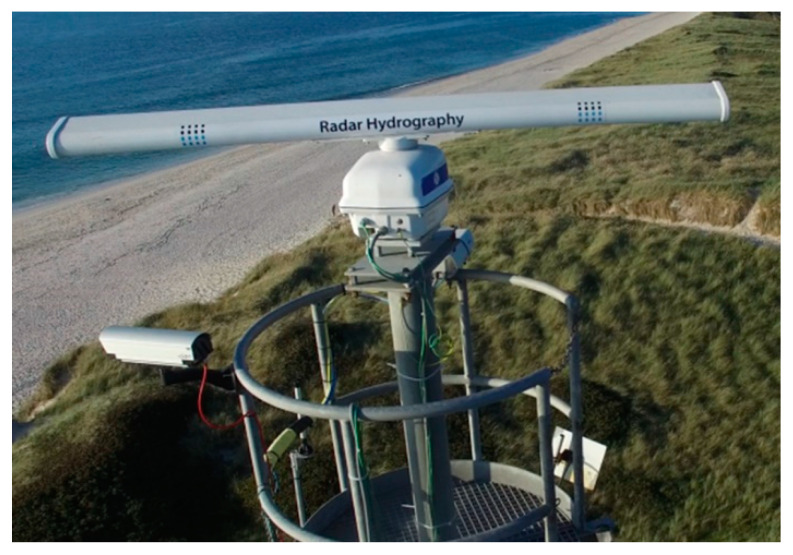
Coastal setup of the coherent on receive marine radar (MR) system on the Island of Sylt in the German Bight of the southern North Sea.

**Figure 2 sensors-21-07828-f002:**
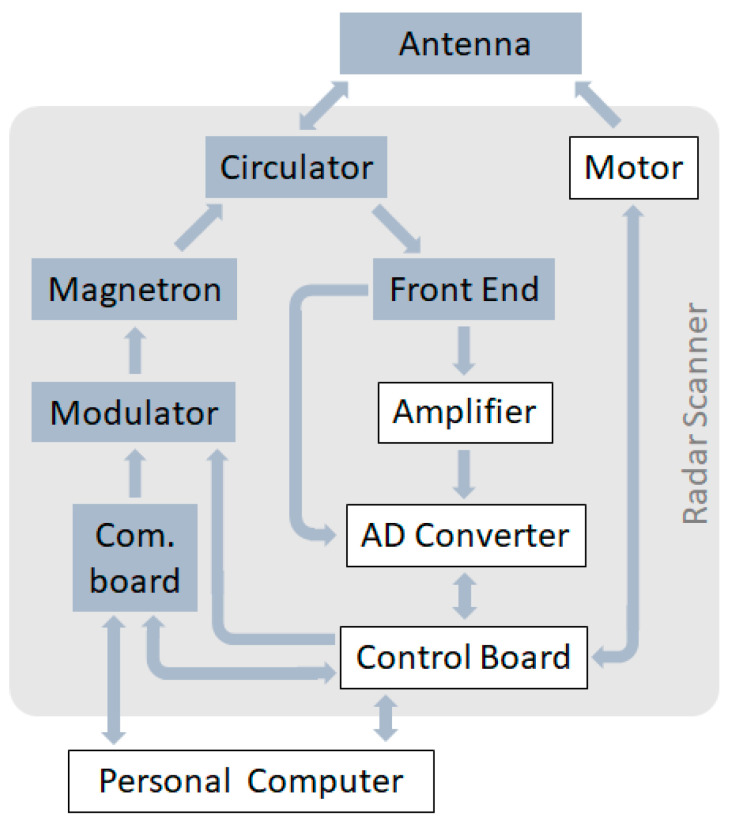
Block diagram of HZG’s coherent on receive MR. All grey shaded boxes are original units from the GEM radar; all white boxes represent custom made add ons to the system.

**Figure 3 sensors-21-07828-f003:**
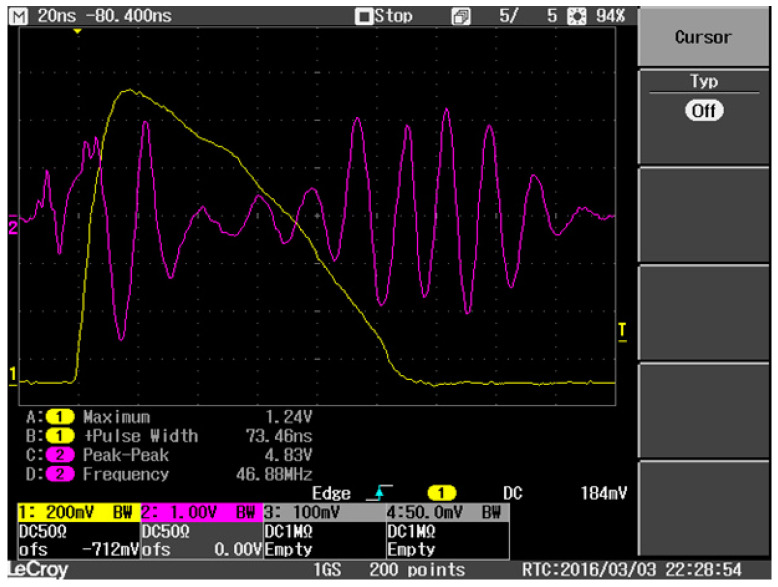
Measurement of the radar pulse (yellow line) and intermediate frequency of the signal at the front end. In this case the pulse width is approximately 73 ns giving an acceptable power and pulse shape.

**Figure 4 sensors-21-07828-f004:**
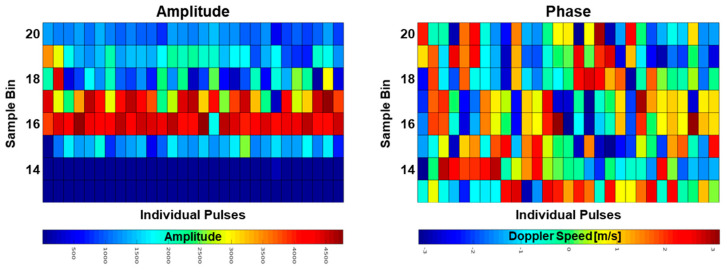
Amplitude (**left**) and phase (**right**) retrieved from the digitized signal. The amplitude is highest where the main energy of the pulse is sampled by the ADC, in this case at bin 16.

**Figure 5 sensors-21-07828-f005:**
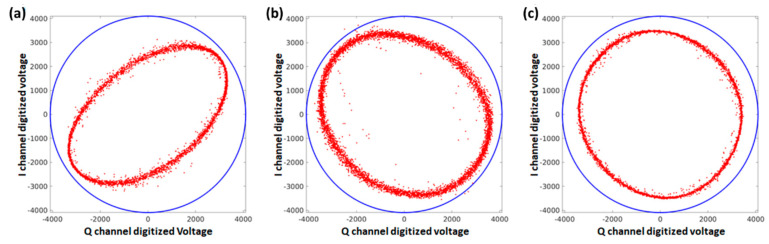
Q versus I channel for a too low (**a**), too high (**b**) and a well tuned (**c**) intermediate frequency within the radar front end.

**Figure 6 sensors-21-07828-f006:**
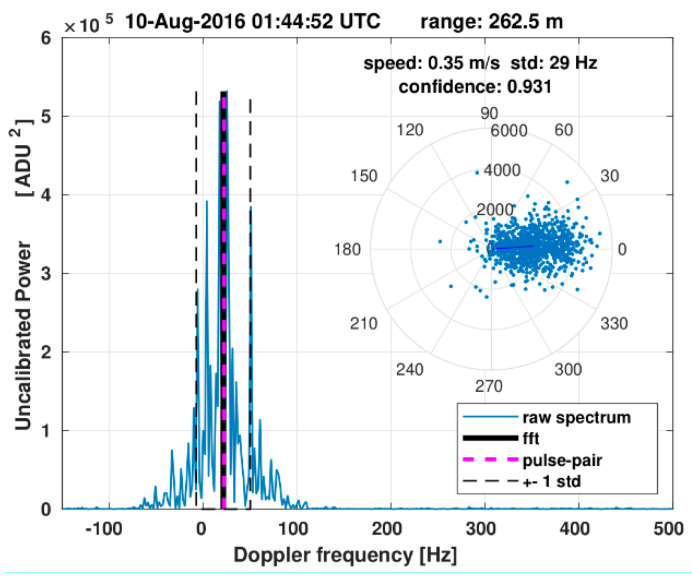
Example Doppler spectrum with integrated Doppler frequency from the first moment (solid black) and pulse-pair (dashed magenta) method. Also shown are the polar representations of the pulse-pair phase differences.

**Figure 7 sensors-21-07828-f007:**
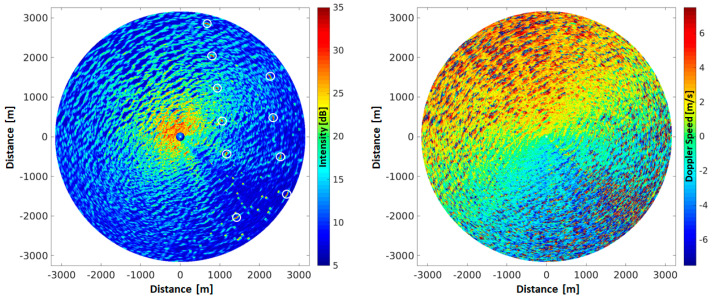
Radar intensity (**left**) and Doppler speed (**right**) image resulting from one single antenna rotation (at 0.6 Hz) from the MR operated at the Research Platform FINO-3, located in the German Bight of the North Sea. The data were collected on 29 October 2017 at 11:54 UTC utilizing horizontal (HH) polarization.

**Figure 8 sensors-21-07828-f008:**
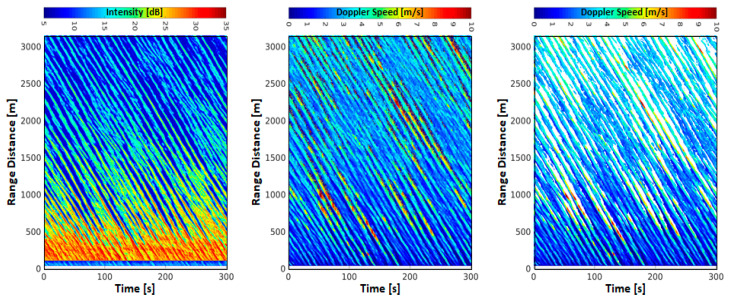
Radar data acquired at FINO-3 on 29 October 2017 from 12:13 UTC on for 5 min. The radar was operated in the static mode with horizontal polarization and the antenna oriented into the peak wave direction of 315°. Radar intensity (**right**), Doppler speed (**center**) and Doppler speeds with areas below a confidence level (Equation (5)) of 0.5 were masked white (**left**). Doppler speeds were retrieved using a 1000 pulses corresponding to 0.5 s.

**Figure 9 sensors-21-07828-f009:**
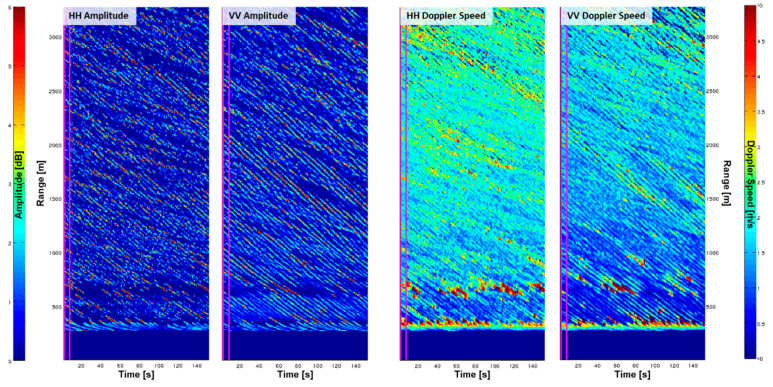
Intensity and Doppler speeds retrieved from the MR with a fixed antenna pointing towards west at the site of Sylt, Germany. The data were collected on the 13 March 2014 with a HH-polarized antenna at 17:00 UTC and VV-polarized antenna at 18:00 UTC using the same radar scanner.

**Figure 10 sensors-21-07828-f010:**
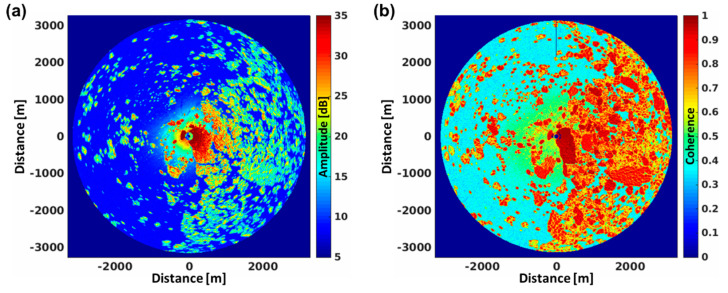
Marine radar measurements of sea ice at VV-polarization in the Fram Strait, acquired from the RV Lance on 3 September 2015 at 14:00 UTC. At the time of acquisition, the vessel was moored to the huge ice float to the east. The amplitude (**a**) and coherence (**b**) were retrieved from 50 rotations (100 s).

**Table 1 sensors-21-07828-t001:** Main parameters of the coherent on receive radar scanner. All components used from the original scanner unit are marked in grey boxes.

Peak Power (nominal)	12 kW
Transmit frequency	9410 ± 30 MHz
Pulse width	60 ± 20 ns
Pulse repetition frequency	2 or 1 kHz
Intermediate frequency	60 ± 2 MHz
Intermediate frequency bandwidth	20 MHz with ±10% tolerance
Noise figure (radar front end)	Nominal 3.5 dB
Antenna rotation speed	up to 36 rpm
Linear amplification	30 to 40 dB
Digitization frequency	80 MHz, 2 channel with 14 bit
Angular transmitter resolution	0.012

**Table 2 sensors-21-07828-t002:** Main characteristics of the radar antennas from GEM.

Length	4′	7.5′	9′
Horizontal beamwidth	1.8° ± 0.1°	1.2° ± 0.1°	0.85° ± 0.1°
Vertical beamwidth	22° ± 2°	22° ± 2°	22° ± 2°
Sidelobes within ±10°	−26 dB	−25 dB	−24 dB
Sidelobes outside ±10°	−30 dB	−30 dB	−30 dB
Gain	≥27 dBi	≥30 dBi	≥31 dBi

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
