# Peer review of "A Coherent on Receive X-Band Marine Radar for Ocean Observations"

_sensors, 2021, doi:10.3390/s21237828_

Round 1

Reviewer 1 Report

In the present manuscript, Horstmann and collaborators present the application of a coherent-on-receive marine radar to estimating wave derived quantities. The articles is clearly organized and well written, and aims at presenting the radar architecture and design, as well as some of its results. Considering it has been submitted to a Special Issue of Sensors, focusing on the applications of marine radars to the ocean environment, the article appears to suit well the scope of the issue. 

That being said, there are two aspects that strike to me as absent: First, as a matter of internal consistency, I miss presenting a comparison or further assessment of the parameters obtained by the system. For instance, by having a measure of the dominant wave period and wave height, it could be possible to assess whether the range of speeds as derived are consistent with orbital velocities, for instance. This could enable to evaluate system performance and capabilities.

The second, and perhaps more fundamental, is that the authors appear to omit previous developments of similar systems. At least, they are not present in the literature review. For instance, Trizna and collaborators have a few publications, although just in conference proceedings, dating as far back as 2010 (e.g. https://doi.org/10.1109/OCEANS.2010.5663917 and https://doi.org/10.23919/OCEANS.2013.6741112, to name just two). Other works, including one this year, also have developed their own COR systems (https://doi.org/10.1109/OCEANSE.2017.8084883, https://doi.org/10.1029/2020RS007173). While I acknowledge the authors do not claim novelty,  I think it is required to highlight the differences between this system and these others.

Therefore, I recommend a revision of the article. Whether the novelty, or lack of, is sufficient to be published, I  defer the decision to the editor.

Specific Points

Figure 3: a) I think showing the mean value for each sample bin among the pulses. When I eyeballed the figure, I thought sample bin 15 was the one to be selected. Also, can the variance among pulses be considered in any way, as secondary selection criterion?

Figure 3b) the phases show a large variability for individual sample bins. How representative is the mean in this case? Or is the phase estimated using other metric?

L201: "where" instead of "were"
Eq 5: Is this equation written using an Einstein convention. I assume there is a sum over j. If that is the case, I guess an j=1 to something (no limit is provided) ought to appear in the denominator as well?
Also, why the limits 1 and 511? Are those fixed or can they be modified?
Fig 6 and text: "Example" should be "Example".
L293: Is it really more accurate? Averaging over a larger number of samples would smooth the results and yield less variability among bins, but does not necessarily make it more accurate.
L293: The use of 0.5 s averaging must be related to the dominant phase speed of the waves. Since we don't know environmental wave properties it is difficult to know whether waves change shape significantly over that time window and smear the results.

Fig 6: I don't follow why Doppler speeds are all positive. Especially at shorter ranges, the backscatter intensity suggests no shadowing is taking place. Hence, we must be seeing the troughs of the waves, and therefore negative speeds (away from the antenna) ought to be present. 

Author Response

For our response to reviewer 1 please refer to the uploaded PDF file.

Reviewer 2 Report

Excellent paper!

Author Response

We would like to thank reviewer 2 and are very happy to hear that he liked the paper!

Reviewer 3 Report

The paper presents a dopplerized marine radar and provides some examples of its activity in the context of METOC applications. This work resembles the works of Dennis Trizna, but there are no references to them. Only the work of 1985 is given. There are also digital Doppler radars that do not require improvements in coherence. Such radars are also used for METOC applications, for example [A.V. Ermoshkin, and I. A. Kapustin, Estimation of the wind-driven wave spectrum using a high spatial resolution coherent radar, Russ. J. Earth Sci., 19, ES1005, doi:10.2205/2019ES000662, 2019.]

In addition, it is necessary to clarify:

What is the rotation speed of the antenna in Figure 7?

Typo in Figure 9 – VV Doppler Speed?

What was the speed of the RV Lance at the time of receiving Figure 10?

Author Response

Please refer to uploaded pdf-file.

Reviewer 4 Report

Comments of “A Coherent on Receive X-Band Marine Radar for Ocean Observations” by Jochen Horstmann et al.

A description of HZG's MR system with the involved tuning and processing steps as well as with intensity and Doppler measurement examples has been presented in the manuscript. It is interesting. However, the content presented in the manuscript likes a technical report. No theoretical contribution has been given. Considering the new marine radar system operating in the coherent on receive mode, I suggest to add more signal processing details in the manuscript.

A minor suggestion: change “exemple” to “example” in the caption of Figure 6.

Author Response

Please refer to uploaded pdf-file.

Round 2

Reviewer 1 Report

I would like to thank the authors for the revision and response. I think they address my previous observations.

Perhaps I would like to see the response they provide in the rebuttal, regarding the positive-only Doppler speeds seen in Fig. 8 to be transferred somewhat in the text as I think is relevant for interpreting system results.

Other than that, I consider it can be published as is. I would not need to review it again.

Reviewer 4 Report

I satisfy their response.